# mTOR Pathway Substrates Present High Activation in Vascular Malformations and Significantly Decrease with Age

**DOI:** 10.3390/diagnostics14010038

**Published:** 2023-12-25

**Authors:** Jakub Kopeć, Elżbieta Sałacińska-Łoś, Magdalena Orzechowska, Monika Sokolnicka, Aleksandra Gawłowska-Marciniak, Przewratil Przemysław

**Affiliations:** 1Pediatric Surgery and Oncology Department, Medical University of Łódź, 90-419 Lodz, Poland; emk.losie@wp.pl (E.S.-Ł.); agawlowska@mnc.pl (A.G.-M.); przemyslaw.przewratil@umed.lodz.pl (P.P.); 2Molecular Carcinogenesis Department, Medical University of Łódź, 90-752 Lodz, Poland; magdalena.orzechowska@umed.lodz.pl; 3Pathomorphology Department, Medical University of Łódź, 90-752 Lodz, Poland; monika.sokolnicka@umed.lodz.pl

**Keywords:** sirolimus, rapamycin, vascular anomalies, lymphatic malformation, venus malformation, mTOR

## Abstract

Background: Vascular anomalies often result in aesthetic flaws, pain, and impair the quality of life. They require challenging treatments that frequently do not provide the desired results. The mammalian target of rapamycin (mTOR) is directly involved in the development of these malformations. However, the exact mechanism behind mTOR dysregulation has not been unambiguously defined. The purpose of this study is to investigate the activation of selected substrates of mTOR to partially assess its involvement in the disease process. Methods: We analyzed tissue samples collected from patients with vascular anomalies treated in our department. We included patients with histopathological diagnoses of lymphatic, venous, capillary malformations, mixed lesions, and a control group of healthy skin samples. We stained the samples using H and E and immunohistochemistry. We used primary antibodies against p70 S6 Kinase, 4EBP1, and p-4EBP1. We graded their color reactions. The statistical analyses were performed using the FactoMineR and factoextra R v.4.1 packages. *p*-values < 0.05 were considered statistically significant. Results: The analysis of 82 patients showed that healthy tissue vessels expressed lower levels of tested mTOR pathway substrates compared to high activation in vascular malformations. Elevated substrate expression in a comparison between sexes revealed higher P-4EBP1 expression in the female malformation group. We observed a decrease in mTOR substrate expression with age. Conclusion: The higher expression of mTOR substrates in vascular malformations compared to healthy tissue confirms their involvement in abnormal vascular development. Age-related changes in mTOR substrate expression highlight the need for timely intervention. Our study contributes to the understanding of the mTOR signaling pathway in vascular malformations and highlights its potential as a therapeutic target, contributing to personalized medicine.

## 1. Introduction

Vascular anomalies (VAs) are the most common lesions present at birth and result from abnormal embryonic vascular development. They comprise a heterogeneous group of congenital vascular malformations and tumors, which are classified based on the nature of vascular flow and the type of vessels involved. According to the International Society for the Study of Vascular Anomalies (ISSVA), VAs can be categorized into vascular malformations (VMs) such as capillary, venous, and lymphatic malformations, and vascular tumors (Table 1).

Most VMs are visible at birth, while a minority of cases may become evident later in life. They exhibit growth patterns that coincide with the child’s development and may spontaneously enlarge during infections. These anomalies often cause cosmetic defects, but in some cases, they can lead to more severe conditions. Therefore, an accurate diagnosis, preceded by radiological investigations such as ultrasound, CT scan, and MRI, is crucial for appropriate treatment. VMs can cause pain and significantly impact the affected patient’s quality of life. Among them, complex and mixed vascular malformations pose the greatest challenge in terms of treatment. They are frequently large, resulting in deformities, chronic pain, and coagulopathies. Moreover, these lesions can become life-threatening by impairing the function of adjacent organs. Unfortunately, the available treatment options for such cases are limited. The current standard of treatment involves sclerotherapy/ablation combined with surgical resection, and less frequently, laser therapy [1]. However, these methods may prove insufficient for extensive lesions in difficult and atypical locations. In recent years, inhibitors of the mechanistic target of rapamycin (mTOR) pathway have emerged as an effective therapeutic option for such patients.

Lymphatic malformations (LMs) are associated with mutations of the *PIK3CA* (phosphatidylinositol-4,5-bisphosphate 3-kinase catalytic alpha subunit) gene, while Kaposiform Hemangioendothelioma (KHE) is associated with the GNA14 [2] (G protein alpha subunit 14) mutation [1,3,4]. Genetic testing has become the basis for diagnosis, and repurposing PI3K pathway inhibitors can be employed as a new strategy for the treatment of these diseases. The oncogenic mutation of *PIK3CA*, which is involved in upstream mTOR signaling, promotes the mTOR pathway and contributes to the development of a subset of vascular malformations, including LMs and VMs. Therefore, the expression of the mTOR signaling pathway may play a key role in the pathogenesis of lymphatic anomalies. Despite the differences in nature between LM, Kaposiform Lymphangiomatosis (KLA), and KHE, treatment with mTOR inhibitors, such as rapamycin, has shown positive responses.

The mTOR metabolic pathway (Figure 1) regulates cell growth and proliferation, and enhances VEGF expression [5], thereby contributing to the regulation of angiogenesis and lymphangiogenesis. The dysregulation of the mTOR pathway is believed to be one of the probable causes for the formation of vascular malformations, as its products are directly involved in vascular development. Sirolimus, an mTOR inhibitor, has been proven to be an effective treatment for challenging lymphatic anomalies [1,6,7,8]. Although the exact mechanism by which rapamycin attenuates mTOR signaling is not fully understood, recent studies have suggested its potential effectiveness in venous and mixed lesions [9]. It is thought that the rapamycin-FKBP12 complex inhibits mTOR function by disrupting the interaction between Raptor and mTOR, thus interrupting mTORC1 connections to specific substrates [10].

The use of sirolimus and its antiproliferative properties, which enable the inhibition of the mTOR pathway, directly halt, reverse, or minimize the effects of vascular anomalies. Sirolimus is a well-tolerated drug that is easy to administer and rarely causes adverse effects [11]. These findings are supported by published observations from leading centers worldwide, including the Cincinnati Children’s Hospital Medical Center [11], Boston Children’s Hospital and Harvard Medical School [11], Hospital Universitario La Paz, Madrid, Spain [12], and many Japanese centers [13]. The number of patients treated at these major centers already exceeds hundreds, with their numbers steadily increasing, which demonstrates the safety and efficacy of this method.

The efficacy of mTOR inhibitors indirectly suggests the deregulation of the mTOR pathway in vascular malformations and increased expression of certain key proteins involved in this metabolic pathway’s function. mTOR exists in two protein complexes: mTORC1 (rapamycin-sensitive) and mTORC2 (rapamycin-resistant). The phosphorylation of known mTOR products, such as eukaryotic translation initiation factor binding protein 4E (4EBP1) and ribosomal protein kinase S6 1 (S6K1), which mediate cell growth and protein synthesis, including lymphangiogenesis, is regulated by mTORC1. The overexpression of these proteins in a patient may suggest early eligibility for mTOR blocker therapy to prevent relapse and reduce the disease’s progression.

The specificity of mTORC1 complexes towards target substrates S6K1 and 4EBP1, and similarly, mTORC2 towards Akt makes the study of their expression in cells a potentially reliable tool for assessing mTOR kinase activity in the disease process. In this study, we investigated the activation of S6K and 4EBP1, known substrates of mTORC1, to assess mTOR kinase activity in the disease process.

## 2. Materials and Methods

### 2.1. Patient Characteristics

We analyzed tissue samples collected from patients with vascular anomalies treated in our department during the years 2012–2020. We included patients with histopathological diagnoses of lymphatic, venous, and capillary malformations, and mixed lesions: venous-lymphatic malformations, capillary-lymphatic malformations, and venous-capillary malformation. We included preauricular appendages as examples of healthy skin tissue. All anomalies were classified according to the ISSVA classification systems. The specimens were obtained by biopsy, curettage, or resection. This study was approved by the Ethical Review Board of the Medical University of Lodz, (No. RNN/307/19/KE) and was performed in accordance with the Committee guidelines and regulations. The study is consistent with the Declaration of Helsinki ethical principles.

### 2.2. Immunohistochemistry for mTOR Pathway Components

All tissue samples were fixed with 10% formalin, routinely embedded in paraffin, cut into 4 μm thick serial sections, and used for hematoxylin and eosin staining and immunohistochemical staining. Immunohistochemical staining was performed using a PowerVision system from Immunologic. We used the primary antibodies against p70 S6 Kinase (49D7) Rabbit mAb (2708S), 4E-BP1 (53H11) Rabbit mAb (9644S) and Phospho- 4E-BP1 (Thr37/46) (236B4) Rabbit Mab (2855S) from Cell Signaling Technology. Immunohistochemical staining was scored by two independent doctors (JK and MS—pathologists). Samples assessed the percentage (from 0 to 100%) of endothelial cells of vessels showing a positive reaction with antibodies. A positive reaction was considered a strong color reaction observed in at least 1% of endothelial cells). The assessment was made in the Olympus BX41 light microscope at 100× magnification in ten fields of view with the largest number of vessels.

### 2.3. Statistical Analysis

The data collected were summarized with descriptive statistics including median with ranges. Due to the non-parametric distribution of continuous variables (age, S6K1, 4EBP1, and p-4EBP1), the comparison between clinical subgroups was performed with the two-sample Wilcoxon test and Kruskal–Wallis followed by a post-hoc Wilcoxon test. Additionally, the principal component analysis (PCA) was employed to evaluate the spatial distribution of the patients according to the expression of S6K1, 4EBP1, and p-4EBP1. The analyses were performed using FactoMineR and factoextra R v.4.1 packages. *p* values < 0.05 were considered statistically significant.

## 3. Results

### 3.1. mTOR Pathway Substrates Present High Activation in Vascular Malformations

We analyzed the clinical data of 82 patients with diagnoses of vascular anomalies that are summarized in Table 2. The 82 lesions consisted of LM (n = 25), VM (n = 29), mixed malformations (MM) (n = 15), and capillary malformations (CM) (n = 13). The control group was a healthy skin tissue of preauricular appendages (n = 11).

We analyzed the expression of 3 proteins (4EBP1, p-4EBP1, S6K1) that are known to play important roles in the mTOR pathway. Immunohistochemically, a heterogenous staining pattern of 4EBP1, P-4EBP1, and S6K1 was detected (Figure 2).

Healthy tissue vessels expressed low levels of 4EBP1 (median = 15, range: 7.5–25.5) and S6K1 (median = 15.5, range: 9.5–33.5) (Figure 3). The p-4EBP1 (median = 11, range: 5.5–16) also had a low expression in the control group. These results indicate that the control group of healthy skin tissue physiologically expresses both 4EBP1 and S6K1 and low activation of p-4EBP1. Phosphorylated, activated forms levels of 4EBP1 and S6K1 that take part in malformation creation aren’t normally elevated in healthy tissue but being involved in vascular development are normally present.

In contrast to the control group, tissues of LM, VM, MM, and CM showed high expression of 4EBP1 (LM: median = 24.5, range: 3.4–45; VM: median = 17.5, range: 6.7–34; MM: median = 13.5, range: 7.9–33.5; CM: median = 22, range: 6.5–65.5), p-4EBP1 (LM: median = 15.2, range: 3.2–46; VM: median = 19, range: 1.5–31.5; MM: median = 17.5, range: 8.2–44; CM: median = 16.5, range: 6–44) and S6K1 (LM: median = 27.5, range: 3.1–60.5; VM: median = 24, range: 15.5–41.5; MM: median = 22.5, range: 8.9–50.5; CM: median = 31.5, range: 6.7–49) in nearly all the cases.

Elevated mTOR substrates expression comparison between sexes revealed higher P4ebp1 (males control group: median = 11, range: 6–16; females control group: median = 9.5, range: 5.5–13.5; males malformation group: median = 14, range: 1.5–44; females malformation group: median = 18.75, range: 6–46;) expression in women malformation group.

Having confirmed higher expression of mTOR downregulation substrates we compared the group between sexes. Generally, there is a similar expression of 4EBP1 in both sexes but higher activation of p-4EBP1 in women (Figure 4).

Comparisons between sexes and groups were made. Higher expression levels of 4EBP1 were found in men samples in both the control group and the malformations group (males control group: median = 17.5, range: 13–25.5; males malformation group: median = 21.75, range: 6.5–63.5). Higher expression of activated p-4EBP1 was found in women in the malformations group but the control group expressed higher p-4EBP1 levels in men.

A similar expression of S6K1 levels was found in both sexes in malformation groups but higher for men in control (males control group: median = 16.5, range: 10–33.5; females control group: median = 12.5, range: 9.5–15.5; males malformation group: median = 25, range: 6.7–46; females malformation group: median = 25.75, range: 3.1–60.5) (Figure 5).

### 3.2. mTOR Pathway Substrates Expression Significantly Decreases with Age

Clinically, we observe high activity and progression tendency in younger patients that decrease with age leading to stabilization of the disease. We decided to analyze the elevated mTOR substrate levels compared to the age of our patients and divided them into three distinct age groups comprising patients below 1 year of age (10 patients), 1–12 years of age (65 patients), and above 12 years of age (17 patients) (Figure 6 and Figure 7).

Expression levels were compared between age groups excluding the control group and important observations were made.

We found a considerable decrease in protein expression with increasing age for all proteins in malformation groups excluding control but statistically significant only for 4EBP1 (<1: median = 26.75, range: 14.5–63.5; 1–12: median = 21, range: 3.4–65.5; 12>: median = 14.5, range: 3.4–34). The expression decrease to age trend is clearly visible in the malformation group which we confirm in our clinic’s observations. The statistical significance of only 4EBP1 may be due to the restricted group sizes (Figure 8).

The comparison between the age groups, including the control group (CG), is shown in Figure 9.

Even though the trend of decreasing expression is visible in both the control and VMs it is more expressed in the VM group, suggesting that an ongoing remission of abnormal vascular development processes may be occurring but simultaneously the tissue may become less responsive for possible treatment in time.

### 3.3. There Is a Higher Expression of Activated p-4EBP1 in VMs than in LMs and a Higher Expression of S6K1 and 4EBP1 in Lymphatic Malformations

There is a significant trend toward difference in expression of non-phosphorylated substrates in lymphatic and venous malformations. S6K1 and 4EBP1 showed more expression in lymphatic malformations. However, the phosphorylated substrates expressed higher for venous malformations. These observations may suggest that rapamycin may be useful in selected cases of both lymphatic and venous malformations. However, it should be reconsidered in the future in comparison with mTOR and p-mTOR levels as to our knowledge they may be activated independently. Also, in vivo models may be very useful in determining the sirolimus effect on both kinds of malformations. Rapamycin can inhibit or lessen the abnormal vascular development in various ways preventing positive feedback activation of mTOR or downregulating the VEGF expression. 

### 3.4. Principal Component Analysis Confirms High mTOR Substrate Expression That Decreases with Age

Spatial partitioning analysis was conducted. In PC1/PC2/vascular malformations are characterized by higher expression of all three proteins compared to controls. 4EBP1 and S6K1 are more correlated with each other than with p-4EBP1, although all three proteins interact similarly. The non-activated substrates act more similarly than phosphorylated versions.

A significant decrease in protein expression with increasing age is once more evident.

In PC2/PC3 space there is a cloud overlapping with VM at (0,0) point. This group is ‘neutral’ in terms of the expression of all three proteins. The other types of malformations are differentiated from VMs in the PC2 and PC3 spaces. It can be seen that the control group, most LMs, and MMs are distinct and spatially separated from VMs. 

For this reason, analyses were performed in these groups, as it can be seen that they are internally heterogeneous. Analyses were performed for VM and LM as the most numerous.

In both cases, spatial variation specific to the malformation region was revealed.

This observation was confirmed statistically for S6K1 and p-4EBP1 in VMs. PCA is not equal to classical statistical analysis and tends to present more significant results.

Correlation circles indicate a high correlation between the expression of the mTOR participants as well as the differentiating of individuals along with the *x*-axis (Figure 10).

## 4. Discussion

The mTOR signaling pathway is a crucial intracellular signaling pathway involved in cell growth, metabolism, and apoptosis, which plays a major role in various pathological mechanisms such as lymphangiogenesis, lymphatic metastasis, and cancer development in different types of neoplasms, including colorectal, gastric, liver, breast and uterine cancer [14].

It functions as a central regulator of protein synthesis, cellular energy levels, and nutrient availability, integrating signals from growth factors, nutrients, and cellular stressors.

The mTOR pathway is composed of two distinct protein complexes: mTOR complex 1 (mTORC1) and mTOR complex 2 (mTORC2). These complexes have distinct compositions and functions within the cell. The primary focus of this discussion will be on mTORC1, as it plays a major role in the context of vascular malformations and related pathologies [10,14].

mTORC1 is composed of several proteins, including the mammalian target of rapamycin (mTOR), the regulatory-associated protein of mTOR (Raptor), and mLST8 (also known as GβL). Activation of mTORC1 occurs in response to various signals, including growth factors (such as insulin and insulin-like growth factor 1), amino acids (particularly leucine), and cellular energy levels (such as ATP). Activation of mTORC1 leads to downstream phosphorylation and regulation of multiple targets involved in protein synthesis and cell growth [10,14].

One of the key downstream targets of mTORC1 is the eukaryotic translation initiation factor 4E-binding protein 1 (4EBP1). When mTORC1 is active, it phosphorylates 4EBP1, preventing its binding to eukaryotic translation initiation factor 4E (eIF4E). This phosphorylation event releases eIF4E, allowing it to initiate protein translation and synthesis of ribosomal components necessary for cell growth and proliferation [10,14].

Another important target of mTORC1 is the ribosomal protein S6 kinase 1 (S6K1). Activation of mTORC1 leads to the phosphorylation of S6K1, which in turn phosphorylates multiple downstream targets involved in ribosome biogenesis, protein synthesis, and cell cycle progression.

Dysregulation of the mTOR pathway has been implicated in various pathological conditions, including cancer and vascular malformations. In the context of vascular malformations, the activation of mTORC1 has been observed, leading to aberrant angiogenesis and lymphangiogenesis. This dysregulation can contribute to the development and progression of vascular anomalies, including lymphatic malformations, Kaposiform lymphangiomatosis, and Kaposiform Haemangioendothelioma [14].

Sirolimus, an mTOR inhibitor, exhibits an antiproliferative effect on lymphatic vessels and is clinically utilized for the treatment of challenging lymphatic anomalies. Sirolimus binds to the FK-binding protein 12, forming a complex that hinders the phosphorylation and activation of 4EBP1 and S6K1. The phosphorylation of these proteins stimulates the production of ribosomal components necessary for protein synthesis and cell cycle regulation [9].

In this study, we aimed to investigate the activation of the mTOR pathway in a subset of vascular malformations. mTOR substrates are normally expressed in healthy tissue although their concentration remains low and so is the activation to phosphorylated forms [14]. The expression levels of 4EBP1, p-4EBP1, and S6K1 were found to be significantly higher in vascular anomalies compared to normal lymphatic vessels, indicating the involvement of the mTOR pathway in lymphangiogenesis in patients with lymphatic anomalies and angiogenesis in other vascular malformations. Immunostaining of p-4EBP1 may serve as a useful tool for distinguishing lymphatic anomalies from normal lymphatic vessels. It is important to note that 4EBP1 and S6K1 are downstream targets of mTOR that can be inhibited by rapamycin. 

Although almost most of the staining patterns of the examined samples were similar for the proteins tested, a few deviated from the pattern which may suggest an originally different clinical diagnosis. 

However, the histopathological differentiation based on mTOR protein expression can help diagnose the type of malformation of excised tissue there is an obvious tendency in non-invasive therapies for VMs that do not require surgical excision and therefore tissue sampling [9]. The emergence of targeted gene therapy has introduced new possibilities for diagnosing vascular malformations and offers several advantages. Targeted gene therapy allows for precise targeting of the specific genetic alterations driving the malformation, enabling personalized treatment approaches.

Activating mutations in the *PIK3CA* gene have been identified in various types of vascular malformations, including lymphatic malformations (LMs) and venous malformations (VMs). mTOR inhibitors like sirolimus primarily target the mTOR pathway, they also exert downstream effects on the PI3K pathway. Combining mTOR inhibitors with PI3K pathway inhibitors may offer synergistic effects in targeting the dysregulated pathways involved in these malformations [9,15,16].

By focusing on the molecular pathways and gene mutations implicated in the disease, targeted gene therapy aims to address the root cause of vascular malformation, potentially leading to improved therapeutic outcomes. Traditional treatment options for vascular malformations, such as surgery or sclerotherapy, may provide temporary relief but often have limitations in terms of complete eradication or long-term control of the malformation. Targeted gene therapies, on the other hand, directly modulate the dysregulated signaling pathways and molecular mechanisms involved in the malformation, offering the potential for more effective and sustained therapeutic responses. Therefore, sirolimus and other similar drugs targeting signaling pathways or in other ways offering personalized treatment based on genetic testing may be the future of VM’s therapies. In addition to the *PIK3CA* gene, several other genes have been identified as potential targets for targeted therapies in vascular malformations TIE2 (*TEK*) gene, *GNAQ* and *GNA11*, Notch pathway genes, *KRAS* and *BRAF* genes [2,9,17].

Targeted gene therapies and gene testing in VMs are a new, dynamically evolving field that brings to the table many future benefits. 

Due to difficult supplies and limited stocks of available antibodies during the COVID-19 pandemic, it proved impossible to use p-S6K1-phosphorylated antibody staining in this study. Based on the analysis of the literature, the p-S6K1 protein is present in an activated form at elevated concentrations also in healthy tissues hence we believe that basing part of the conclusions on the relationship between 4-EBP1 and p-4EBP1 may be partially sufficient [14].

The comparison of the expression of mTOR pathway proteins between sexes showed that men had lower expression of p-4EBP1 than women. Although the clinical implications of this difference may not be immediate, it suggests a potential disparity in vessel proliferation activity between males and females, necessitating further research and confirmation.

Additionally, we observed a significant difference in the expression of 4EBP1 among different age groups (excluding the control group). This finding indicates a decline in the expression of mTOR substrates with increasing age, suggesting age-related changes in mTOR pathway protein expression levels. It is worth noting that this observation holds importance irrespective of statistical significance, highlighting the need for further investigation into age-related alterations in mTOR pathway activity.

This can also be seen in the correlations between expression and age expressed numerically.

It is important to mention that our study did not evaluate the expression of mTOR and p-mTOR substrates, as pathway activation leading to malformations does not always involve mTOR directly. Separate and independent activation of 4EBP1 and S6K1 can occur without mTOR involvement, as the phosphorylation of mTOR substrates is reliant on down-regulation and parallel regulation by other kinases and feedback mechanisms [14,18].

Sirolimus, a specific mTOR inhibitor, has the ability to bind to mTOR and downregulate its function. It has been reported to simultaneously inhibit the mTOR and vascular endothelial growth factor (VEGF) pathways in lymphatic malformations, demonstrating efficacy in many cases even those without mTOR activation [1,19]. Both the mTOR and VEGF pathways play crucial roles in lymphangiogenesis. Combining mTOR pathway inhibition with other pathways, such as the RAS or TIE pathways, is being considered to enhance treatment efficacy. 

Although sirolimus is generally effective and safe for most patients with complicated lymphatic anomalies, some individuals do not respond well to the treatment. Further research is needed to determine whether it is possible to predict which patients will benefit from sirolimus therapy and whether the expression levels of mTOR pathway proteins can serve as biomarkers for treatment response. As mentioned before the most promising is targeted gene therapy.

In conclusion, our study investigated the activation levels of mTOR pathway substrates in vascular malformations and explored their relationship with age. The findings support the hypothesis that dysregulation of the mTOR pathway plays a crucial role in the pathogenesis of these anomalies. The age-related decrease in substrate expression emphasizes the importance of early intervention for optimal treatment outcomes. The results also suggest the potential utility of mTOR inhibitors, such as sirolimus, as an effective therapeutic option for vascular malformations. 

The higher expression of mTOR substrates in vascular malformations compared to healthy tissue confirms their involvement in abnormal vascular development. Furthermore, the observed gender differences in substrate expression, with higher levels of activated 4EBP1 in women, suggest potential gender-specific variations in the pathogenesis of these anomalies, warranting further investigation.

Age-related changes in mTOR substrate expression highlight the need for timely intervention, particularly in younger patients, to achieve optimal treatment outcomes. The decline in substrate expression with increasing age may indicate ongoing remission of abnormal vascular development processes, but it also raises the possibility of reduced responsiveness to treatment options over time.

We believe that early diagnosis and appropriate timing of treatment at the earliest stage may be crucial in terms of outcome and prognosis.

The heterogeneity observed within different types of vascular malformations, as evidenced by spatial variation and distinct molecular mechanisms, suggests the necessity of tailored treatment approaches. The identification of specific genetic mutations associated with certain types of malformations and lesions, such as *PIK3CA* mutations in lymphatic malformations and GNA14 mutations in Kaposiform Haemangioendothelioma, has opened doors for targeted therapies using pathway inhibitors [2,9].

While sirolimus has shown efficacy in treating difficult lymphatic anomalies, further research is needed to elucidate the precise mechanisms underlying mTOR pathway dysregulation and to evaluate the efficacy of targeted therapies in different types of malformations. 

In conclusion, our study contributes to the understanding of the mTOR signaling pathway in vascular malformations and highlights its potential as a therapeutic target contributing to personalized medicine. Further research in this area will pave the way for the development of new treatment strategies and better management of vascular anomalies, thus improving patient outcomes by understanding systemic treatment mechanisms.

## Figures and Tables

**Figure 1 diagnostics-14-00038-f001:**
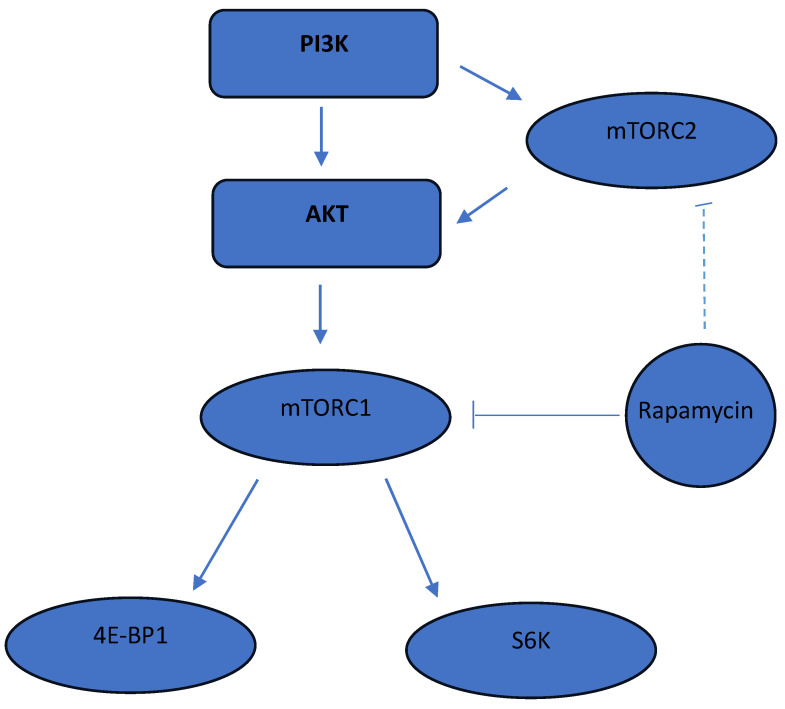
The mTOR signaling pathway.

**Figure 2 diagnostics-14-00038-f002:**
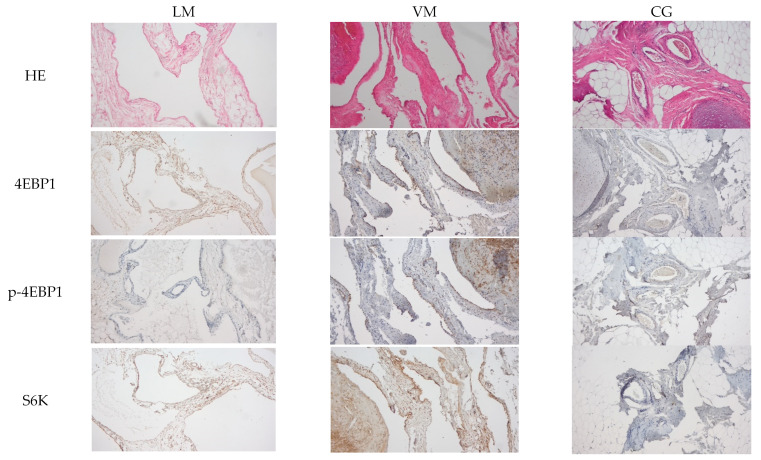
Examples of tissue samples and staining.

**Figure 3 diagnostics-14-00038-f003:**
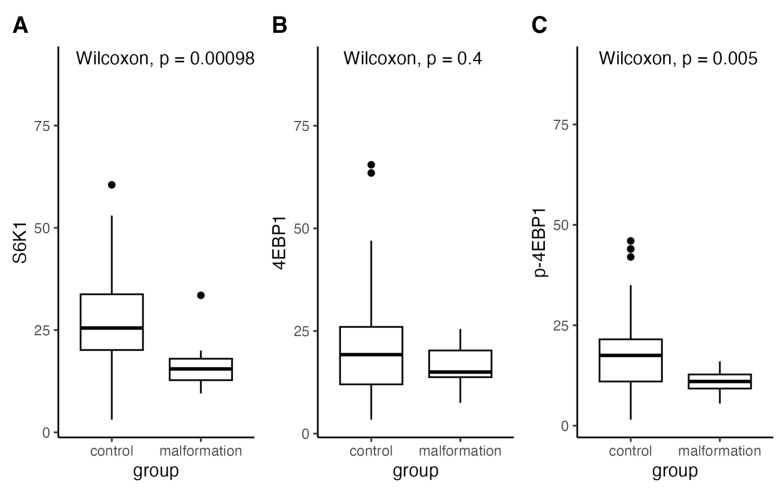
Distribution of expression of S6K1, 4EBP1, and p-4EBP1 in the control and malformation groups compared with Wilcoxon test.

**Figure 4 diagnostics-14-00038-f004:**
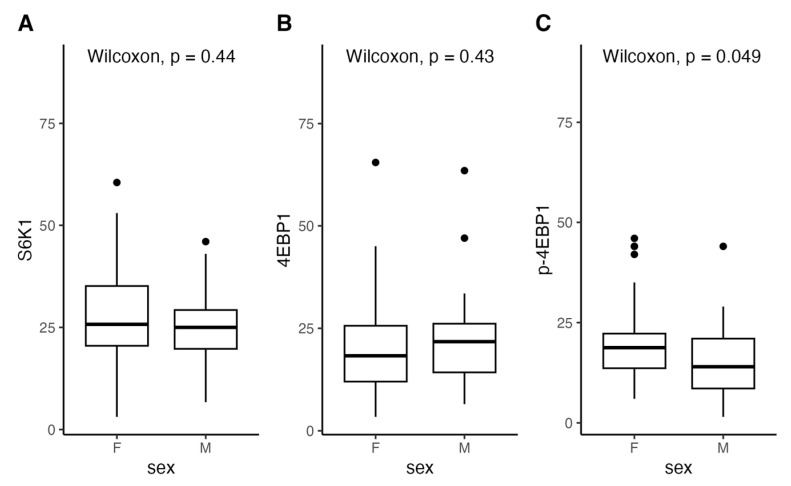
Distribution of expression of S6K1, 4EBP1, and p-4EBP1 according to gender compared with Wilcoxon test.

**Figure 5 diagnostics-14-00038-f005:**
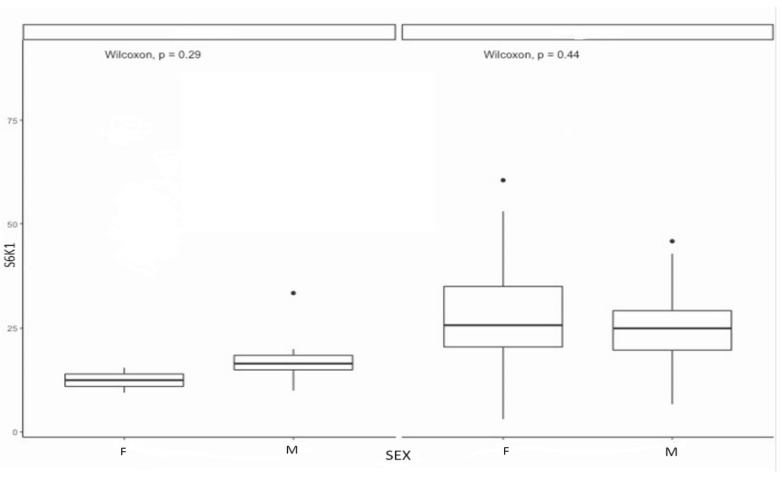
Distribution of expression of S6K1 according to gender.

**Figure 6 diagnostics-14-00038-f006:**
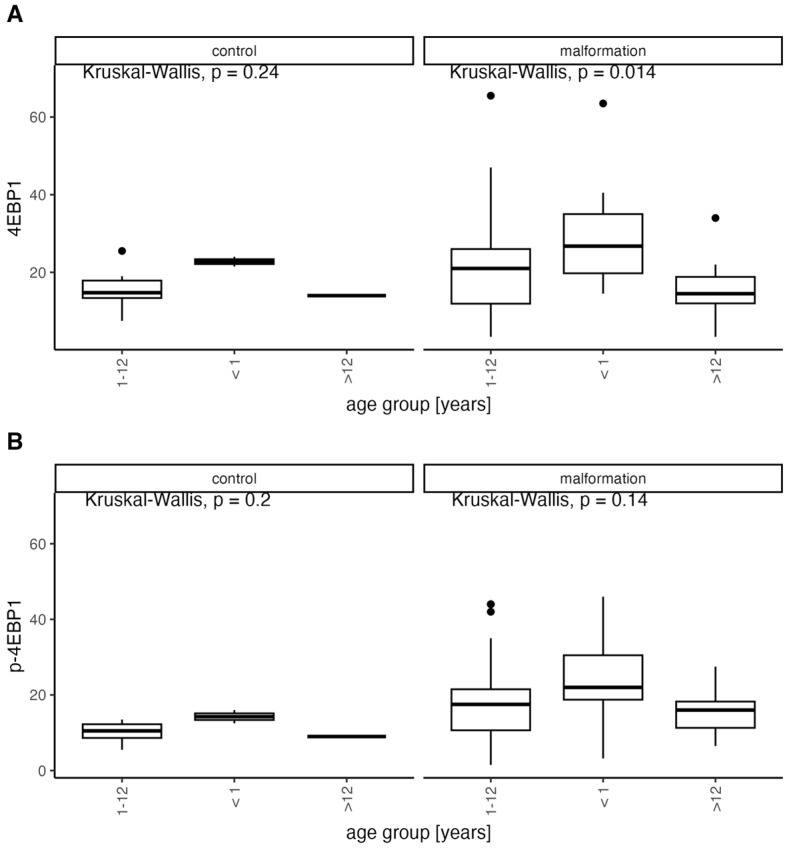
Distribution of expression of 4EBP1 and p-4EBP1 in the control and malformation groups according to the age of onset compared with Kruskal-Wallis test.

**Figure 7 diagnostics-14-00038-f007:**
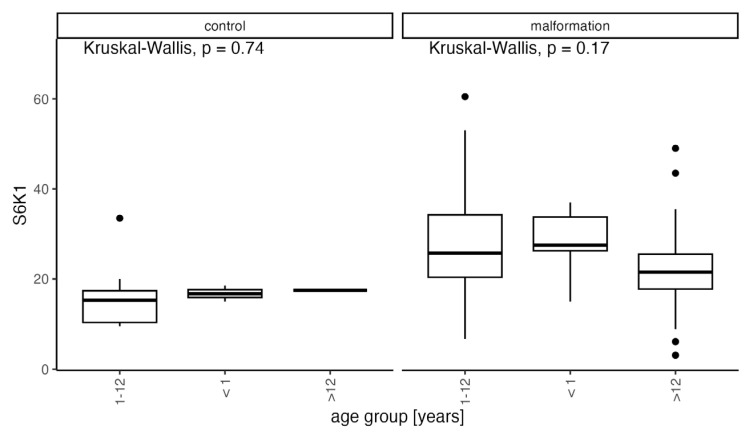
Distribution of expression of S6K1 in the control and malformation groups according to the age of onset compared with Kruskal-Wallis test.

**Figure 8 diagnostics-14-00038-f008:**
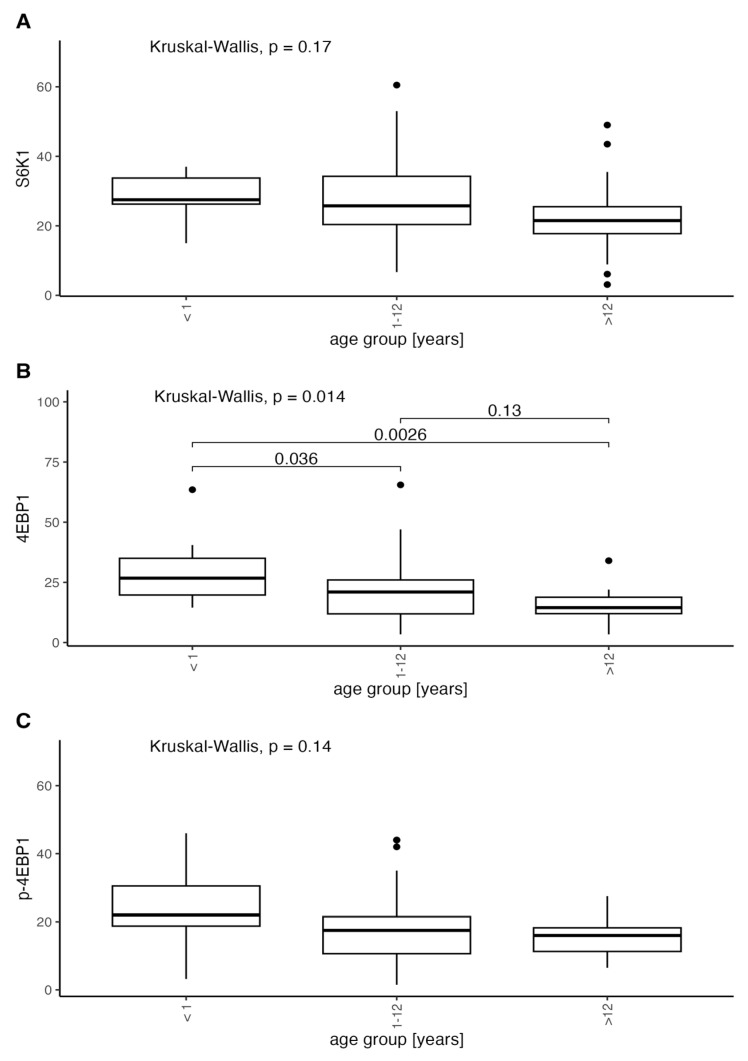
Distribution of expression of S6K1, 4EBP1, and p-4EBP1 according to the age of onset compared withKruskal-Wallis test and post-hoc pairwise Wilcoxon test.

**Figure 9 diagnostics-14-00038-f009:**
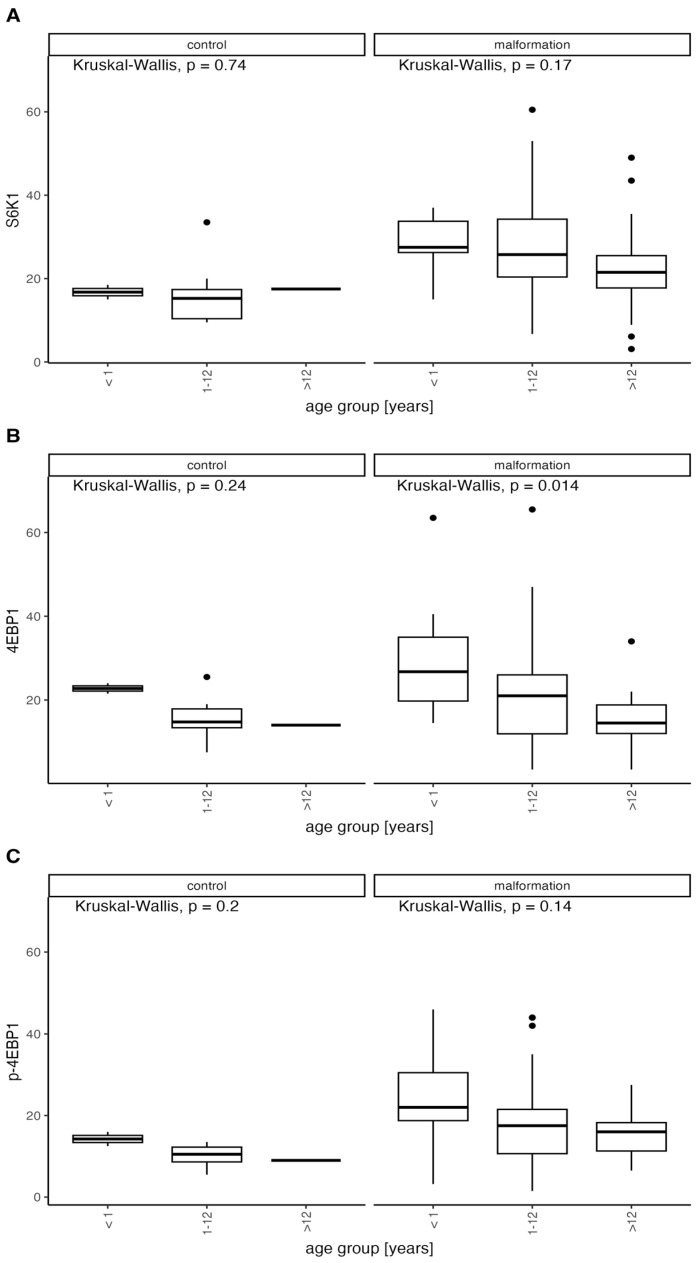
Distribution of expression of p-4EBP1, 4EBP1 and S6K1 in the control and malformation groups according to the age of onset compared with Kruskal-Wallis test.

**Figure 10 diagnostics-14-00038-f010:**
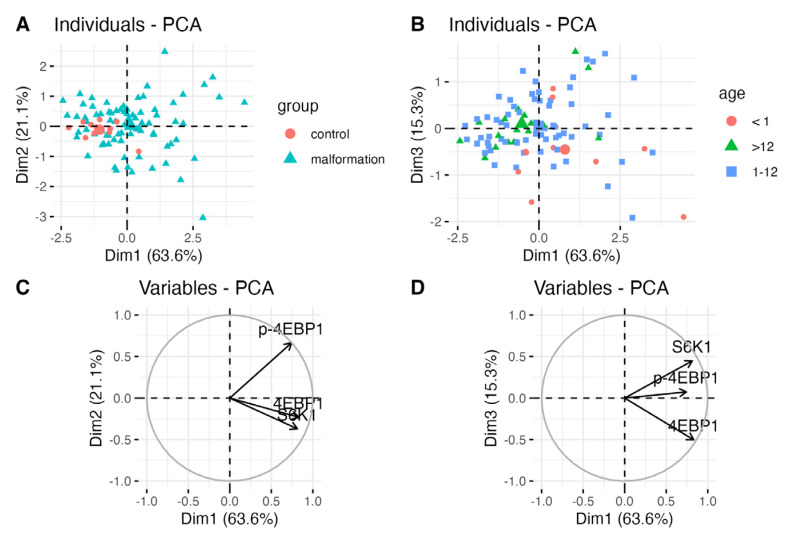
Correlation circles indicate high correlation of the expression of the mTOR participants as well as differentiating of individuals along with *x*-axis.

**Table 1 diagnostics-14-00038-t001:** ISSVA classification for Vascular Malformations.

Vascular malformations ISSVA classification					
Vascular tumors			Vascular malformations		
Benign tumors	Borderline tumors	Malignant tumors	Simple	Combined	Associated with Other Anomalies
Infantile hemangioma - Endothelial cell proliferation - GLUT-1 marker positive	Hemangio-endothelioma	Angio-sarcoma	Venous Malformations (VM) Blue rubber bleb nevus Syndrome Glomovenous Malformation	CM + VM CM + LM CM + AVM LM + VM	Klippel—Trenaunay Syndrome
Congenital hemangioma - Excessive Angiogenesis with Capillary Lobules - GLUT-1 Marker negative - Fully developed at birth	Others	Others	Lymphatic Malformation (LM)Macrocystic Microcystic Mixed Cystic	CM + LM + VM CM + LM + AVM CM + VM + AVM	CLOVES Syndrome
Tufted Angioma			Capillary Malformation (CM) Teleangiectasia Nevus Simplex Others	CM + LM + VM + AVM	Sturge—Weber Syndrome
Spindle Cell Hemangioma			Arterio-Venous Malformation (AVM)		Parkes—Weber Syndrome
Epitheloid Cell Hemangioma - Others			Arterio-Venous Fistula (AVF) Hereditary Hemorrhagic Teleangiectasia (HHT)		Others

Abbreviations: CM—capillary malformations, VM—venous malformations, LM—lymphatic malformations, AVM—arterio-venous malformations, AVF—arterio-venous fistula, HHT—hereditary hemorrhagic teleangiectasia.

**Table 2 diagnostics-14-00038-t002:** Staining pattern with a cut-off point at 5%.

	Control (11)	LM (25)	VM (29)	MM (15)	CM (13)
Staining pattern		+/−	+/−	+/−	+/−
P-4EBP1	+	23/2	27/2	15/0	13/0
4EBP1	+	23/2	29/0	15/0	13/0
S6K1	+	24/1	29/0	15/0	13/0

## Data Availability

The data presented in this study are available on request from the corresponding author. The data are not publicly available due to personal data of patients.

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
