# Peer review of "mTOR Pathway Substrates Present High Activation in Vascular Malformations and Significantly Decrease with Age"

_diagnostics, 2023, doi:10.3390/diagnostics14010038_

Round 1
Reviewer 1 Report
Comments and Suggestions for Authors
Nice paper, well written. Only a minor comment, line 448, Kaposifirm hemangioendothelioma is not a malformation as you mention in page
You often called Kaposi's lymphangiomatosis and Kaposi's hemangioendothelioma. Both are wrong; the correct terms are "kaposiform" lympangiomatosis and "kaposiform" hemangioendothelioma.
Comments on the Quality of English LanguageI think it is acceptable as it is
Author Response
Dear reviewer,
Thank you for your comments and suggestions. I corrected all the Kaposi's to kaposiform. Can't say how theese mistakes slipped through. I also fixed the sentence in line 448 to make it clear that KE is separate in nature from malformations.
Best Regards
Reviewer 2 Report
Comments and Suggestions for Authors
Dear colleagues, it was big interest to read your article. Hope You will work futher with it. We need to define the targets to influence on this malformations grow.
Comments on the Quality of English LanguageAdecvate and understandable language
Author Response
Dear reviewer,
Thank you very much for your interest in our work and the opinion expressed. This is a very important topic for us, as we treat a vast and growing number of patients with vascular leasions in our clinic.
We are already conducting further research, which we hope to publish in the near future.
Best regards.
Reviewer 3 Report
Comments and Suggestions for Authors
This is a very well written article on the subject.
It is fluent and well thought.
Although the findings are specific for a single centre, its contribution focuses on a key aspect of diagnosis and treatment.
Are you considering a multi-centre study to further develop your project?
Finally, only one suggestion:
Page17, line 441-442: it would be more appropriate to say “We believe that early diagnosis and appropriate timing of treatment at the earliest stage may be crucial in terms of outcome and prognosis.”
Comments on the Quality of English LanguageAppropriate use of the English language.
Author Response
Dear reviewer,
Thank you very much for your interest in the article and your opinion.
I have taken the liberty of using the suggestions for the sentence about early inclusion of treatment.
Of course, we want to conduct a multicentre study. I think this can become easier with this publication.
We are also conducting other studies on the topic of vascular lesions, which we hope to publish soon.
Best regards